# *SCGN* deficiency results in colitis susceptibility

Luis F Sifuentes-Dominguez[1], Haiying Li[2], Ernesto Llano[2], Zhe Liu[3], Amika Singla[2], Ashish S Patel[1], Mahesh Kathania[2], Areen Khoury[2], Nicholas Norris[1], Jonathan J Rios[1,4,5], Petro Starokadomskyy[2], Jason Y Park[4,6], Purva Gopal[6], Qi Liu[2], Shuai Tan[2,3], Lillienne Chan[1], Theodora Ross[2], Steven Harrison[7], K Venuprasad[2], Linda A Baker[7], Da Jia[3], Ezra Burstein[2,8]*

[1]Department of Pediatrics, University of Texas Southwestern Medical Center, Dallas, United States; [2]Department of Internal Medicine, University of Texas Southwestern Medical Center, Dallas, United States; [3]Key Laboratory of Birth Defects and Related Diseases of Women and Children, Department of Paediatrics, West China Second University Hospital, State Key Laboratory of Biotherapy and Collaborative Innovation Center of Biotherapy, Sichuan University, Chengdu, China; [4]McDermott Center for Human Growth and Development, University of Texas Southwestern Medical Center, Dallas, United States; [5]Seay Center for Musculoskeletal Research, Texas Scottish Rite Hospital for Children, Dallas, United States; [6]Department of Pathology, University of Texas Southwestern Medical Center, Dallas, United States; [7]Department of Urology, University of Texas Southwestern Medical Center, Dallas, United States; [8]Department of Molecular Biology, University of Texas Southwestern Medical Center, Dallas, United States

*For correspondence:
ezra.burstein@utsouthwestern.edu

Competing interests: The authors declare that no competing interests exist.

**Abstract** Inflammatory bowel disease (IBD) affects 1.5–3.0 million people in the United States. IBD is genetically determined and many common risk alleles have been identified. Yet, a large proportion of genetic predisposition remains unexplained. In this study, we report the identification of an ultra rare missense variant (NM_006998.3:c.230G > A;p.Arg77His) in the *SCGN* gene causing Mendelian early-onset ulcerative colitis. *SCGN* encodes a calcium sensor that is exclusively expressed in neuroendocrine lineages, including enteroendocrine cells and gut neurons. SCGN interacts with the SNARE complex, which is required for vesicle fusion with the plasma membrane. We show that the *SCGN* mutation identified impacted the localization of the SNARE complex partner, SNAP25, leading to impaired hormone release. Finally, we show that mouse models of *Scgn* deficiency recapitulate impaired hormone release and susceptibility to DSS-induced colitis. Altogether, these studies demonstrate that functional deficiency in SCGN can result in intestinal inflammation and implicates the neuroendocrine cellular compartment in IBD.
DOI: https://doi.org/10.7554/eLife.49910.001

## Introduction

Inflammatory bowel disease (IBD) is a condition that results from both genetic predisposition and environmental exposures, which has become increasingly prevalent with the advent of industrialization around the world (*Kaplan, 2015*; *Kaplan and Ng, 2017*). Twin concordance studies were the first to recognize the contribution of genetic factors to IBD pathogenesis (*Brant, 2011*). At the present time, a series of large genome-wide association studies (GWAS) have identified over 200 variants associated with genetic risk (*Liu et al., 2015*). Most of these variants are intergenic and confer only modest risk (with odds ratios typically below 2). Studies in pediatric populations have focused on

monogenic Mendelian traits and have identified over 50 genes that can be linked to early-onset IBD (*Uhlig and Schwerd, 2016*). Interestingly, the vast majority of these mutations result in complex syndromes of immune dysfunction or developmental alterations that resemble only in part the phenotype of adult-onset IBD. In aggregate, the identified loci contribute to about 20–30% of the genetic risk, highlighting the need for additional gene discovery.

The genes implicated in both GWAS and familial studies are enriched for pathways that are critical in host-microbiome interactions. This is likely the case because the luminal aspect of the intestine represents the largest epithelial surface in the body (*Furness et al., 2013*) and houses the largest concentration of human-associated microbiota (*Li et al., 2014a*; *Qin et al., 2010*). As such, it is not surprising that the intestine contains the greatest number of immune cells in the body (*Castro and Arntzen, 1993*), which must preserve a delicate balance between tolerance and protective immunity to achieve normal intestinal function. The main mechanisms mediating the physiologic balance between protective immunity and tolerance are thought to involve the interplay between immune populations of the intestine and the epithelial barrier. In contrast, much less is known about the role that specialized cellular compartments of the intestine, such as enteroendocrine cells (EECs) and gut neurons, may play in intestinal immune homeostasis.

EECs comprise 1% of the intestinal epithelium and in aggregate represent the largest endocrine cell population in the body (*Sternini et al., 2008*). These cells secrete peptide hormones in response to nutrients or other small molecules, which are recognized by specific cell surface receptors often located on the luminal membrane. Most secreted products produced by EECs are involved in regulating the digestive process, gastrointestinal motility, and the metabolic adaptation to nutrient influx (*Drucker, 2007*; *Murphy and Bloom, 2006*; *Rindi et al., 2004*). Based on gene expression profiling, there are more than 10 distinct populations of EECs with distinct hormone secretory characteristics in the small intestine (*Furness et al., 2013*). Notably, human mutations impacting *NEUROG3*, the transcription factor required for EEC differentiation, have been described in patients with chronic diarrhea secondary to malabsorption without an inflammatory component (*Wang et al., 2006*). Another related cellular compartment is comprised of gut neurons, which are aggregated in two distinct locations in the intestinal wall: the submucosal plexus and the myenteric plexus. Derived from neural crest progenitors, these cells are integral to the motility function of the gut, and their participation in other intestinal functions is increasingly recognized. In fact, recently, gut neurons were demonstrated to interconnect with EECs and form a neuro-epithelial circuit that is ultimately connected to the central nervous system (*Kaelberer et al., 2018*). However, our understanding of this circuit's activity in immune regulation is limited and a role for these cells in IBD pathogenesis has not received significant attention.

In this study, we identified a homozygous recessive mutation in the *SCGN* gene causing early-onset ulcerative colitis. *SCGN* encodes Secretagogin, a calcium sensing protein that interacts with the SNARE complex (*Bauer et al., 2011*; *Rogstam et al., 2007*; *Wagner et al., 2000*). The SNARE complex is required for secretory vesicle docking with target membranes (*Jahn and Scheller, 2006*). We show that the disease-causing mutation results in loss of SCGN function and that *Scgn*-deficient mice are predisposed to colitis, highlighting the role of this gene, and more broadly the role of the neuroendocrine intestinal compartment, in intestinal immune homeostasis.

## Results

### A very rare variant in SCGN causes inherited early-onset ulcerative colitis

We identified a consanguineous Hispanic family with three of five children affected by an aggressive form of early onset ulcerative colitis (*Figure 1a*). All three probands were diagnosed by standard clinical, endoscopic, and histologic criteria (*Levine et al., 2014*). Two siblings (P1 and P2) were diagnosed at age eight while the third (P3) was diagnosed at age 6. Siblings P1 and P2 had a severe disease phenotype characterized by pancolonic involvement, anti-TNF treatment failure, ultimately requiring procto-colectomy 9 years and 9 months after diagnosis, respectively (*Figure 1b* and *Figure 1—figure supplement 1*). Sibling P3 had a milder phenotype with exclusive rectal involvement, controlled on oral mesalamine up until last point of follow up.

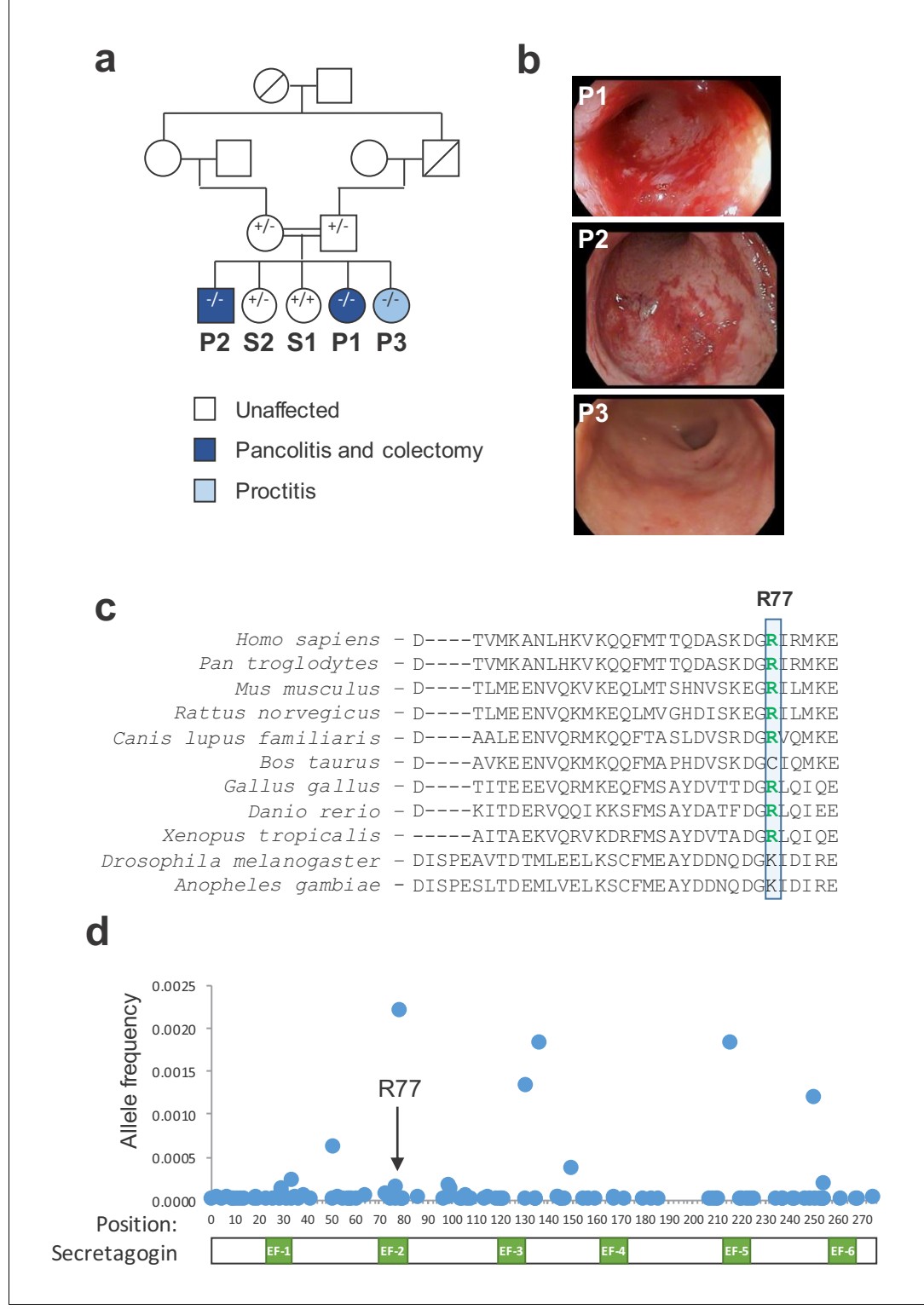

**Figure 1.** A mutation in *SCGN* is linked to early-onset ulcerative colitis. (**a**) Pedigree of index family. Probands (P1, P2, and P3) and their siblings (S1 and S2) are indicated. *SCGN* genotypes are noted (+ = WT allele, - = R77H allele) (**b**) Representative endoscopic images of the rectum from affected individuals. (**c**) Multispecies alignment of SCGN protein sequences is shown; the residue affected in the rare coding variant found in affected patients (R77) is indicated. (**d**) Frequencies of allele variants of *SCGN* found in ExAC are plotted along the SCGN protein sequence, with the location of the six EF-hands also indicated. R77 is noted by an arrow.

*Figure 1 continued on next page*

*Figure 1 continued*

DOI: https://doi.org/10.7554/eLife.49910.002

The following source data and figure supplements are available for figure 1:

**Source data 1.** Source data for *Figure 1D*.

DOI: https://doi.org/10.7554/eLife.49910.005

**Figure supplement 1.** Clinical course of affected probands.

DOI: https://doi.org/10.7554/eLife.49910.003

**Figure supplement 2.** Areas of shared LOH in P1 and P2 as analyzed by WGS.

DOI: https://doi.org/10.7554/eLife.49910.004

The consanguineous nature of the family pedigree suggested an autosomal recessive trait, where each of the affected individuals inherited a recessive allele from one of their parents. Multiple genomic approaches were used to identify candidate recessive gene variants. Genome-wide microarray genotyping of all siblings was utilized to identify segments of loss-of-heterozygosity (LOH) and copy number variants (*Supplementary file 1*). Next, we sorted these finding along the genome to identify segments of shared changes in the three probands (P1, P2, P3) that were not shared by their unaffected siblings (S1, S2). Through this analysis, we identified large segments of LOH on chromosomes 6 and 12 that were shared between all three affected probands and not present in the unaffected siblings (*Table 1*). Whole genome sequencing of two of the probands confirmed these areas of LOH and indicated that they have more continuity than suggested by the SNP array analysis (*Figure 1—figure supplement 2*). Next, we performed whole-exome sequencing (WES) of all five siblings, and we filtered for rare homozygous nonsynonymous variants present among all the three probands and not their unaffected siblings. WES analysis identified a single rare (MAF <1%) homozygous missense variant in the *SCGN* gene (rs376721140; NM_006998.3:c.230G > A; p.Arg77His) that was shared by the three probands and not their unaffected siblings. This variant mapped to one of the segments of shared LOH on chromosome 6. To ensure additional candidate variants were not missed in areas of low WES coverage within the proband-shared LOH regions, we performed whole-genome sequencing (WGS) of two of the affected probands (P1 and P2). WGS confirmed that the *SCGN* variant is the only rare homozygous nonsynonymous candidate variant in the segments of shared LOH among

**Table 1.** Areas of shared loss-of-heterozygosity (LOH) among affected probands as defined by SNP array.

|        | Start      | End        |
|--------|------------|------------|
| Chr 12 | 71,016,157 | 72,070,710 |
|        | 72,320,251 | 73,669,855 |
|        | 73,671,276 | 75,891,939 |
|        | 77,249,400 | 79,204,389 |
|        | 79,302,419 | 82,139,004 |
|        | 82,147,487 | 83,157,896 |
|        | 83,541,966 | 88,441,286 |
|        | 88,594,159 | 89,910,070 |
|        | 89,915,484 | 91,765,486 |
|        | 91,766,720 | 93,152,115 |
|        | 93,251,750 | 95,410,845 |
| Chr 6  | 16,893,011 | 18,222,277 |
|        | 18,262,607 | 19,682,599 |
|        | 19,686,123 | 19,803,768 |
|        | 19,804,188 | 21,778,105 |
|        | 23,584,375 | 26,148,311 |

DOI: https://doi.org/10.7554/eLife.49910.006

these probands. Finally, familial segregation of the *SCGN* p.R77H mutation was confirmed by Sanger sequencing, showing that both parents were heterozygote carriers, and the unaffected siblings were either a heterozygote carrier or had reference alleles, as per WES results (*Figure 1a*).

The allelic frequency for the variant found in affected individuals in this family is very rare (all ExAC = 0.014%; Hispanics only = 0.017%). As Hispanics are generally underrepresented among public genomic databases, we genotyped 2,000 Hispanic individuals (mostly of Mexican origin) from the Dallas Heart study (*Victor et al., 2004*). The mutation frequency among Hispanics in the Dallas Heart Study was 0.025%. Furthermore, in agreement with the rarity of this allele, no homozygous individual were identified within the Dallas Heart Study nor reported in public genomic databases (ExAC; *Lek et al., 2016*), 1000 Genomes Project (*1000 Genomes Project Consortium et al., 2015*), dbSNP (*National Center for Biotechnology Information, 2019*) or IBD exomes browser (*Inflammatory Bowel Disease Exomes Browser, 2016*).

Further screening of different databases (ExAC, ClinVar) indicates that there are no individuals identified thus far with predicted homozygous essential loss of function variant (stop codon/frameshift). The only potentially deleterious homozygous variant found was a nucleotide deletion (c.83-1delG) predicted to affect a splice acceptor site 5' upstream of exon 2 (rs60502981). This allele is enriched among individuals of African descent, with an allele frequency of 2.8% in this population: four homozygous individuals (all of African descent) are reported in ExAC (0.08% prevalence). Importantly, ClinVar does not include this variant and validation of its predicted effect on splicing or gene function is currently lacking. In fact, all variants of *SCGN* reported in ClinVar are large copy number variants involving dozens to hundreds of genes, with phenotypes associated with congenital anomalies and intellectual disabilities (Accession numbers VCV000608768, VCV000608767, VCV000608764, VCV000443497, VCV000443496, VCV000155430, VCV000150044, VCV000149747).

Secretagogin, the protein encoded by the *SCGN* gene, is a calcium sensing protein predicted to have 6 EF–hand domains (*Wagner et al., 2000*). The identified variant (p.R77H) maps to a highly conserved Arginine in the predicted second EF hand of the protein, which is substituted for Histidine (*Figure 1c*). While this change may be a conservative amino acid substitution, the rarity of this particular allele suggested that it may have functional consequences. To further understand the potential functional importance of the amino acid in question, we considered the mutational burden of the affected site. We collected allele frequencies for all coding variants in *SCGN* noted in ExAC, and mapped them along the length of the encoded protein and its EF hands (*Figure 1d*). We observed only a few residues with significant allelic heterogeneity, typically outside the EF hands. Arginine 77 exhibited limited coding variation, consistent with a potentially important functional role for this residue.

## SCGN is expressed in EECs and intestinal neurons

To gain insight into any possible mechanism connecting *SCGN* to IBD pathogenesis, we first sought to characterize the expression pattern of *SCGN* in the intestine. We performed immunohistochemistry of rectal biopsies from healthy individuals and observed strong expression among scattered triangularly-shaped epithelial cells, morphologically compatible with EECs (*Figure 2a*). When colonic tissues from the probands (P1, P2 and P3) were stained, the pattern and morphology of these cells was not affected (*Figure 2b* and *Figure 2—figure supplement 1*). We next examined in more detail the pattern of SCGN protein expression in the intestine of normal mice. We observed that SCGN was confined to rare cells that co-stained in some instances for Chromogranin A (CGA), a marker of certain EEC populations (*Figure 2c*), with the degree of costaining being greater in the mouse colon, and minimal in the mouse small bowel (*Figure 2d*). Similar findings were made in human colonic tissues, where in a subpopulation of cells, SCGN co-localized with other EEC markers such as Chromogranin B (CGB), Serotonin (5-HT) and glucagon (GCG) (*Figure 2—figure supplement 2*). These findings suggested that SCGN marks a unique sub-population of EECs, in agreement with recent single-cell expression profiling of the murine small intestinal epithelium, which identified *Scgn* as a marker of a specific small intestinal EEC population (*Haber et al., 2017*). In addition, distinct clusters of SCGN positive cells in the submucosal and muscle layers of the murine intestine also co-stained for the neuronal markers TUJ1, SNAP25 and synaptophysin (SYP), consistent with *SCGN* expression in subpopulations of gut neurons (*Figure 2e*, *Figure 2—figure supplement 3*). In contrast with the partial overlap seen in intestinal tissues between SCGN and EEC or neuronal markers, SCGN protein expression was uniform in murine pancreatic islets where it colocalized with CGA (*Figure 2f*).

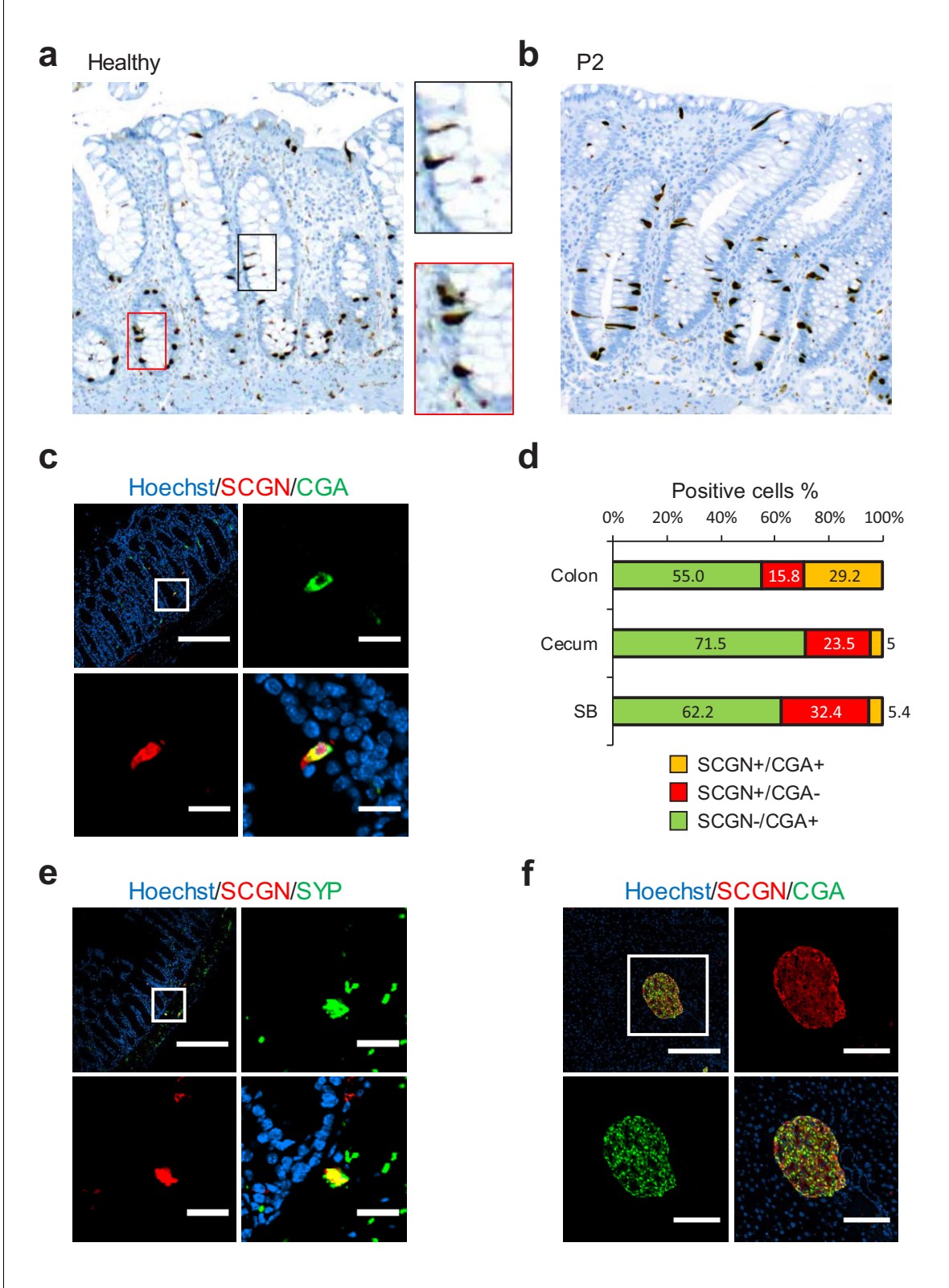

**Figure 2.** SCGN is expressed neuroendocrine cells. (a) Immunohistochemistry (IHC) staining for SCGN in rectal biopsies obtained from a healthy individual. 40x magnification. (b) IHC staining for SCGN in rectal biopsies obtained from P2, one of the probands in the index family. 40x magnification. (c) SCGN and chromogranin A (CGA) immunofluorescence staining of murine colonic epithelium (scale bar 200 µm – inset 20 µm). (d) Morphometric quantification of SCGN and CGA staining patterns in epithelial cells of three sites of the murine gut. Cell population percentages are shown. SB: Small

*Figure 2 continued on next page*

Figure 2 continued

bowel (e) SCGN and synaptophysin (SYP) immunofluorescence staining in subepithelial cells of the murine large intestine (scale bar 200 μm – inset 20 μm). (f) SCGN and CGA immunofluorescence staining in murine pancreatic islets (scale bar 200 μm - inset 100 μm).

DOI: https://doi.org/10.7554/eLife.49910.007

The following source data and figure supplements are available for figure 2:

**Source data 1.** Source data for *Figure 2D*.
DOI: https://doi.org/10.7554/eLife.49910.011
**Figure supplement 1.** Immunohistochemistry (IHC) staining for SCGN in colonic biopsies obtained from probands P1, P2, and P3. 40x magnification.
DOI: https://doi.org/10.7554/eLife.49910.008
**Figure supplement 2.** Immunofluorescence staining for markers of EEC lineage in healthy human colon (scale bar 200μ - inset 40 μm).
DOI: https://doi.org/10.7554/eLife.49910.009
**Figure supplement 3.** Immunofluorescence staining for neuronal markers in murine colon of C57BL/6 mice (scale bar 200 μm - inset 50 μm).
DOI: https://doi.org/10.7554/eLife.49910.010

## SCGN$^{R77H}$ is functionally impaired

EECs secrete peptide hormones upon nutrient stimulation in a process akin to neurotransmitter release. Both nutrient stimulation and neurotransmitter release activities require components of the SNARE complex (*Li et al., 2014b*; *Wheeler et al., 2017*), which mediates membrane fusion events. Previous studies have identified that SCGN interacts with the SNARE complex components SNAP23 and SNAP25, suggesting that this factor regulates neurotransmitter and hormone release in cells that express SCGN (*Bauer et al., 2011*; *Rogstam et al., 2007*). To assess SCGN function in exocytic release, we developed a cellular model of stimulated hormone release using the SCGN-expressing murine enteroendocrine cell line STC-1, which secretes GLP-1 in response to nutrient stimuli (*Hirasawa et al., 2005*; *McCarthy et al., 2015*; *McLaughlin et al., 1998*). Using CRISPR/Cas9 technology to target the first exon of murine *Scgn*, we generated two independent *Scgn* knockout (KO) clones that displayed loss of protein expression (*Figure 3—figure supplement 1*). Next, we measured GLP-1 release in response to docosahexaenoic acid (DHA) exposure, a fatty acid known to stimulate GLP-1 secretion in these cells (*Hirasawa et al., 2005*). Lack of SCGN expression led to a significant decline in GLP-1 secretion when compared to parental cell lines, indicating that SCGN is required for optimal DHA-stimulated GLP-1 secretion from cultured EECs (*Figure 3a*). SCGN expression in KO cell lines was restored using lentiviral vectors encoding human *SCGN*, including the wild-type (WT) or the p.R77H variant (*Figure 3—figure supplement 1*). DHA stimulated GLP-1 secretion was significantly impaired in cells expressing SCGN$^{R77H}$ compared to the isogenic wild-type control cells (*Figure 3b*), suggesting that the disease-associated variant is a hypomorphic allele that leads to altered secretion dynamics in EECs.

To understand the mechanisms that result in dysfunction of hormone secretion, we investigated whether SCGN$^{R77H}$ had impaired binding to SNAP25, its partner in the SNARE complex. Immuno-precipitation of SCGN from STC-1 cells expressing either WT or the R77H variants led to comparable co-precipitation of endogenous SNAP25 from these cells (*Figure 3—figure supplement 2*). Next, we examined whether SCGN$^{R77H}$ might display dysfunctional regulation of SNAP25 subcellular localization. SNAP25 localization in STC-1 cells expressing WT or SCGN$^{R77H}$ proteins was examined by immunofluorescence staining. In parental STC-1 cells, both SCGN and SNAP25 displayed vesicular and plasma membrane localization (*Figure 3c*, left column). This is in agreement with the observation that membrane association of SNARE proteins is required for their biological activity in secretory processes (*Gonelle-Gispert et al., 2000*). Interestingly, the membranous localization of SNAP25 was lost in *Scgn* KO cells (*Figure 3c*, EV lane), and this was restored upon re-expression of SCGN$^{WT}$ (*Figure 3c*), indicating that normal cellular localization of SNAP25 is dependent on the expression of its partner SCGN. In contrast, SCGN$^{R77H}$ failed to restore SNAP25 plasma membrane localization (*Figure 3c*, right column), and these distribution changes were highly significant after image quantification (*Figure 3d*). Consistent with its role in directing membrane localization of SNAP25, SCGN$^{R77H}$ itself also failed to localize to the plasma membrane. In aggregate, these observations indicate that SCGN$^{R77H}$ displays abnormal subcellular localization which also impact its partner, SNAP25, ultimately resulting in altered SNARE function and vesicular secretion.

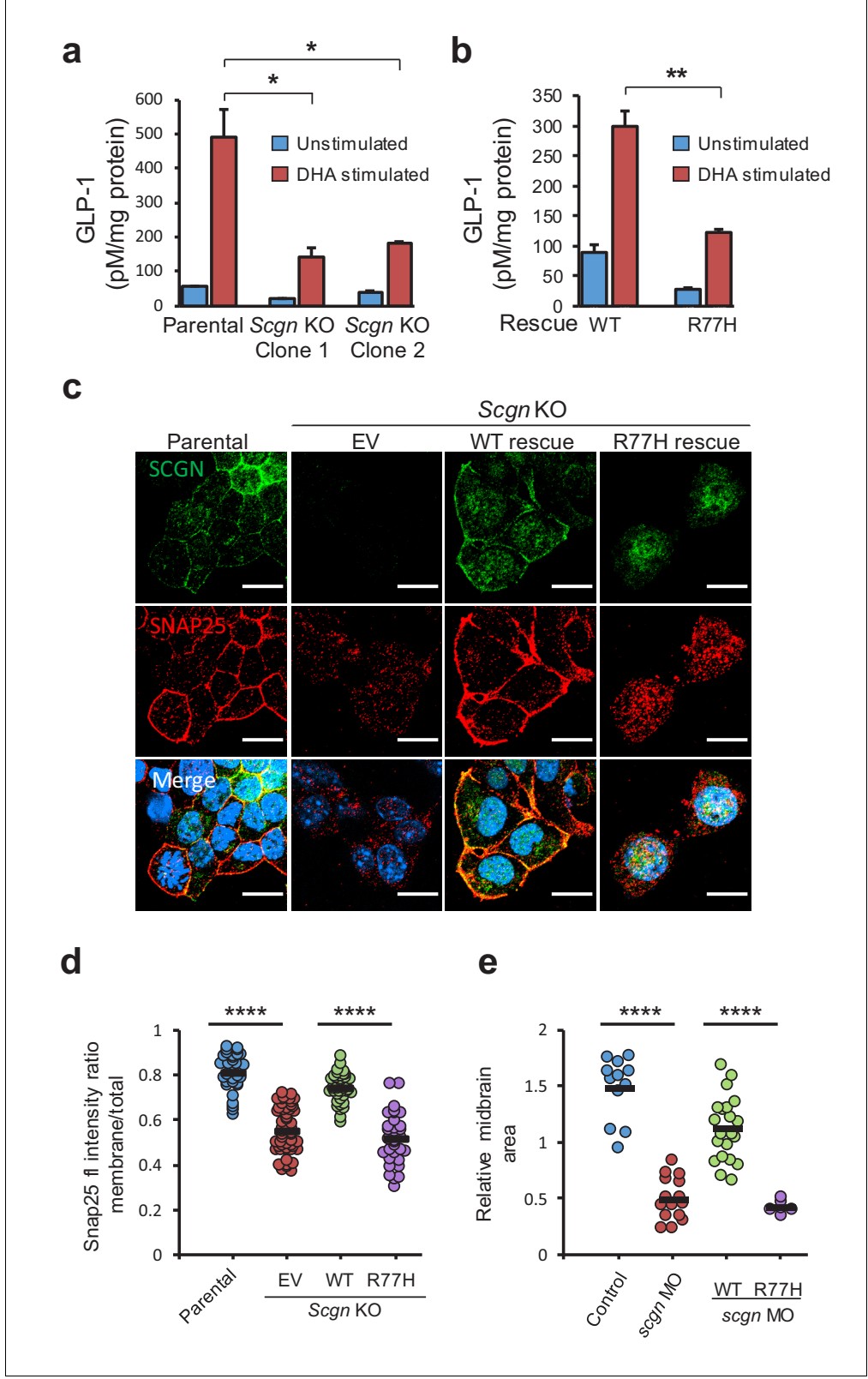

**Figure 3.** *SCGN* p.R77H is a hypomorphic allele. (a) Basal and DHA fatty acid induced GLP-1 release from parental and *Scgn* deleted (KO) clones. GLP-1 values are normalized to total protein content. (b) Basal and DHA stimulated GLP-1 secretion from rescue clones expressing human SCGN[WT] or SCGN[R77H]. (c) Immunofluorescence images showing subcellular localization of SNAP25 and SCGN in parental STC-1 cells and the indicated SCGN KO

*Figure 3 continued on next page*

*Figure 3 continued*

and rescue cell lines. Scale bar 15 µm. (**d**) For the experiment depicted in (**c**), SNAP25 staining intensity ratio between the membranous compartment and the total cellular signal was plotted in the indicated cell lines. Dots indicate individual cells, horizontal bars correspond to the mean within each group. (**e**) Midbrain size of zebrafish after *scgn* targeting with morpholinos or rescue with human SCGN^WT or SCGN^R77H. Dots indicate individual embryos, horizontal bars correspond to the mean within each group. The zebrafish experiments were performed in triplicate. *p<0.05, **p<0.01, ****p<0.0001 unpaired student *t* test in (**a**), (**b**) and (**d**). ****p<0.0001 multiple comparison ANOVA in (**e**). Error bars in (**a**) and (**b**) represent the S.E.M.
DOI: https://doi.org/10.7554/eLife.49910.012

The following source data and figure supplements are available for figure 3:

**Source data 1.** Source data for *Figure 3A*.
DOI: https://doi.org/10.7554/eLife.49910.016
**Source data 2.** Source data for *Figure 3B*.
DOI: https://doi.org/10.7554/eLife.49910.017
**Source data 3.** Source data for *Figure 3D*.
DOI: https://doi.org/10.7554/eLife.49910.018
**Source data 4.** Source data for *Figure 3E*.
DOI: https://doi.org/10.7554/eLife.49910.019
**Figure supplement 1.** *Scgn* deficient clones of STC-1 cells were generated by CRISPR/Cas9 technology.
DOI: https://doi.org/10.7554/eLife.49910.013
**Figure supplement 2.** SNAP25 co-precipitation from SCGN rescue cell lines.
DOI: https://doi.org/10.7554/eLife.49910.014
**Figure supplement 3.** Bright-field images of zebrafish after morpholino injections (MO).
DOI: https://doi.org/10.7554/eLife.49910.015

To determine whether *SCGN* p.R77H also behaved as a loss of function allele in vivo, we investigated the function of SCGN in zebrafish, whose ortholog is 73% identical to the human protein. The most notable phenotype of SCGN loss in this animal model is abnormal midbrain development resulting in significant reduction in the size of this organ (*Deciphering Developmental Disorders Study, 2015*). SCGN suppression using a translation blocking morpholino (MO) targeting the ATG start codon led to the expected developmental phenotype of reduced midbrain size (*Figure 3e*, *Figure 3—figure supplement 3*), visualized using in situ hybridization for HuC (elavl3), an early pan-neuronal marker (*Kim et al., 1996*). Importantly, this phenotype was rescued when human SCGN^WT mRNA was co-injected, whereas co-injection of mRNA encoding the R77H mutant failed to rescue the phenotype. Taken together, these findings are consistent with SCGN^R77H being devoid of normal activity in cellular and in vivo developmental models.

## Scgn-deficient mice are prone to colitis

Given the identification of *SCGN* p.R77H in a pedigree with early onset ulcerative colitis, we next wished to address the role of this gene in colitis development. To that end, we engineered a *Scgn* knockout (*Scgn^-/-*) mouse model by targeting the third exon of murine *Scgn* using CRISPR/Cas9 technology. Multiple pronuclear injections of Cas9 mRNA and custom synthetic gRNA led to numerous independent C57BL/6J *Scgn* targeted founder mice harboring different indels within the third exon of *Scgn*. Two independent founder lineages were selected (*Secret1* and *Secret2*), each with *Scgn* deletions that could be easily detected by PCR-based genotyping (*Figure 4—figure supplement 1a,b*). Founders were backcrossed to wild-type C57BL/6J mice for at least three generations to dilute any possible off-target effects. Homozygous *Secret1* and *Secret2* mice expressed a truncated transcript skipping exon 3 of the gene (*Figure 4—figure supplement 1c,d*). Immunofluorescence staining and imaging of pancreatic islets using an antibody to an epitope not affected by the observed exon skipping demonstrated loss of SCGN protein expression in these two mouse lines (*Figure 4—figure supplement 1e*).

Through heterozygote mating, littermate wild-type (WT) and *Scgn* deficient mice were generated and cohoused in a specific-pathogen-free (SPF) environment. At baseline, *Scgn* deficient mice (*Secret1* and *Secret2*) did not develop any overt phenotype, including no obvious architectural abnormality in the colonic epithelium as seen in HE or alcian blue staining (*Figure 4a*), and no colitis

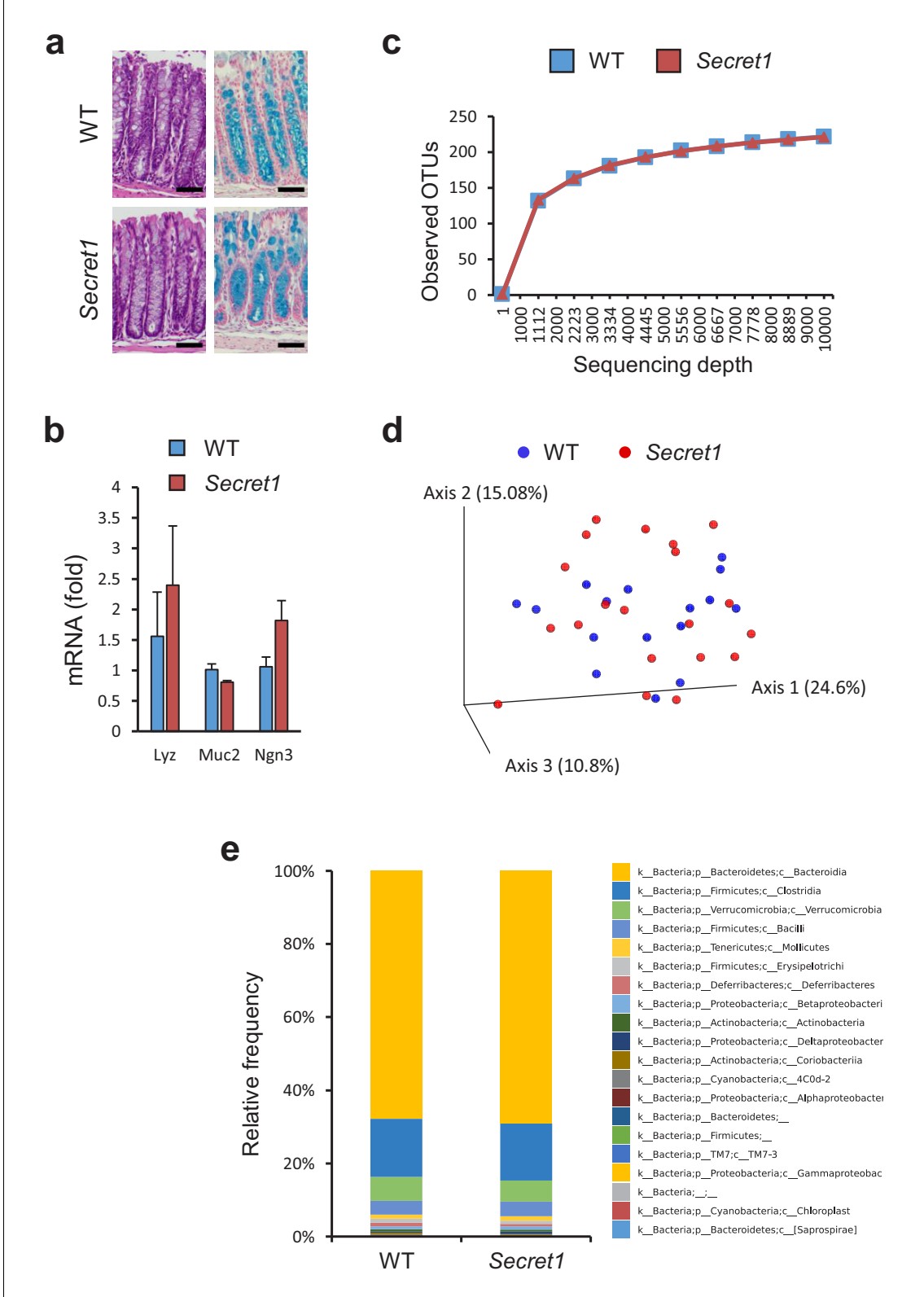

**Figure 4.** *Scgn* deficient mice display intact mucosal architecture and microbiota at baseline. (**a**) Representative colon histologic images from *Secret1* and WT animals under untreated conditions. HE is shown on the left, alcian blue staining on the right. Scale bar 100 μm. (**b**) Baseline expression of prototypical intestinal epithelial cell lineage-specific markers from small bowel in *Secret1* and WT mice was determined by qRT-PCR (n = 3 in each group). (**c**) Microbiome alpha diversity as measured by observed OTU mean counts of fecal 16 s rRNA sequencing. (**d**) Beta diversity by unweighted

*Figure 4 continued on next page*

*Figure 4 continued*

UNIFRAC principal coordinate analysis (PCoA) of fecal 16 s rRNA sequencing. (e) Class level taxonomic composition for stool 16 s sequencing. WT n = 16 *Secret1* n = 20. S.E.M. was used for error bars in (b).

DOI: https://doi.org/10.7554/eLife.49910.020

The following source data and figure supplement are available for figure 4:

**Source data 1.** Source data for *Figure 4B*.
DOI: https://doi.org/10.7554/eLife.49910.022
**Source data 2.** Source data for *Figure 4C*.
DOI: https://doi.org/10.7554/eLife.49910.023
**Source data 3.** Source data for *Figure 4E*.
DOI: https://doi.org/10.7554/eLife.49910.024
**Figure supplement 1.** Engineering *Scgn* deficient animals.
DOI: https://doi.org/10.7554/eLife.49910.021

was noted clinically or histologically even in mice as old as 330 days. Gene expression analyses indicated that key markers of specialized epithelial populations remained intact in these mice (*Figure 4b*). Furthermore, bacterial diversity and microbiota composition were not significantly different between WT and homozygous *Secret1* mice (*Figure 4c–e*).

Next, we examined the susceptibility of *Scgn*-deficient mice to DSS-induced colitis, a widely used model of acute colitis in mice. Importantly, most genetic defects thought to lead to increased IBD susceptibility in humans also display colitis sensitivity in this model (*Perše and Cerar, 2012*). Upon administration of DSS, homozygous *Secret1* male mice experienced worse weight loss (*Figure 5a*), elevated disease activity index (*Figure 5b*), greater mortality (*Figure 5c*), and increased tissue damage by histopathology (*Figure 5d,e*). Greater weight loss was also recapitulated when performing these experiments without gender selection (*Figure 5—figure supplement 1a*) and was also seen in the *Secret2* line (*Figure 5—figure supplement 1b*). In agreement with the more severe colitis in *Scgn* knockout mice, loss of *Scgn* led to increased expression of pro-inflammatory genes in colonic tissues in *Secret1* (*Figure 5f*) and *Secret2* (*Figure 5—figure supplement 1c*) homozygous mice. In aggregate, the data indicate that loss of *Scgn* in mice results in increased susceptibility to colitis.

We examined whether loss of *Scgn* led to appreciable colonic lamina propria leukocyte abnormalities at baseline that could account for the susceptibility of this model to colitis. Flow cytometric analysis of colonic and small bowel lamina propria leukocyte populations from WT and *Secret1* animals showed no differences in populations of cell types examined (*Figure 5—figure supplement 2*).

## Colonic EECs do not modulate DSS sensitivity

Next, we sought to determine if disrupted colonic EEC function or dysfunction of the neuronal compartment was responsible for the effects of SCGN deficiency on intestinal inflammation. To that end, we used a genetic strategy to ablate colonic EECs through deletion of *Neurog3*, a transcription factor required for lineage differentiation (*Mellitzer et al., 2010*). *Neurog3* deletion in the colonic epithelium was accomplished using a colonocyte-specific transgene (*Hinoi et al., 2007*), CDX2-Cre (*Figure 6a*). The resulting mice lost all CGA positive cells in the colonic epithelium, while showing no changes in the small intestine (*Figure 6b*). Gene expression of EEC-specific genes such as *Neurog3* and *Gcg* was lost from colonocytes, while goblet cell markers were unaffected (*Figure 6c*). Importantly, ablation of EECs in the colon led to loss of *Scgn* expression in this compartment as would be expected (*Figure 6c*). Moreover, RNA sequencing analysis from colonocyte isolates also indicated that colonic *Neurog3* deficiency was associated with loss of predicted or known EEC transcripts, including *Scgn* (*Figure 6d*). Functionally, colons from these animals exhibited drastic loss of hormone secretion at baseline, as measured by ex-vivo colonic GLP-1 and GLP-2 secretion in organ cultures. Furthermore, DHA stimulation led to GLP-1 and GLP-2 release from EEC sufficient (*Neurog3*[f/f]) colon explants, but not from colons of EEC ablated (*Neurog3*[ΔCol]) littermates. In addition, GLP-1 and GLP-2 release in response to DHA was completely blunted in colon explants from *Secret1* homozygous animals, consistent with the role of SCGN in this response (*Figure 6e*). Importantly, EEC deficient mice did not have increased susceptibility to DSS-induced colitis (*Figure 6f,g*) suggesting that colonic EECs do not mediate the immunomodulatory effects of SCGN and raising the possibility that SCGN-expressing enteric neurons may modulate intestinal immune homeostasis.

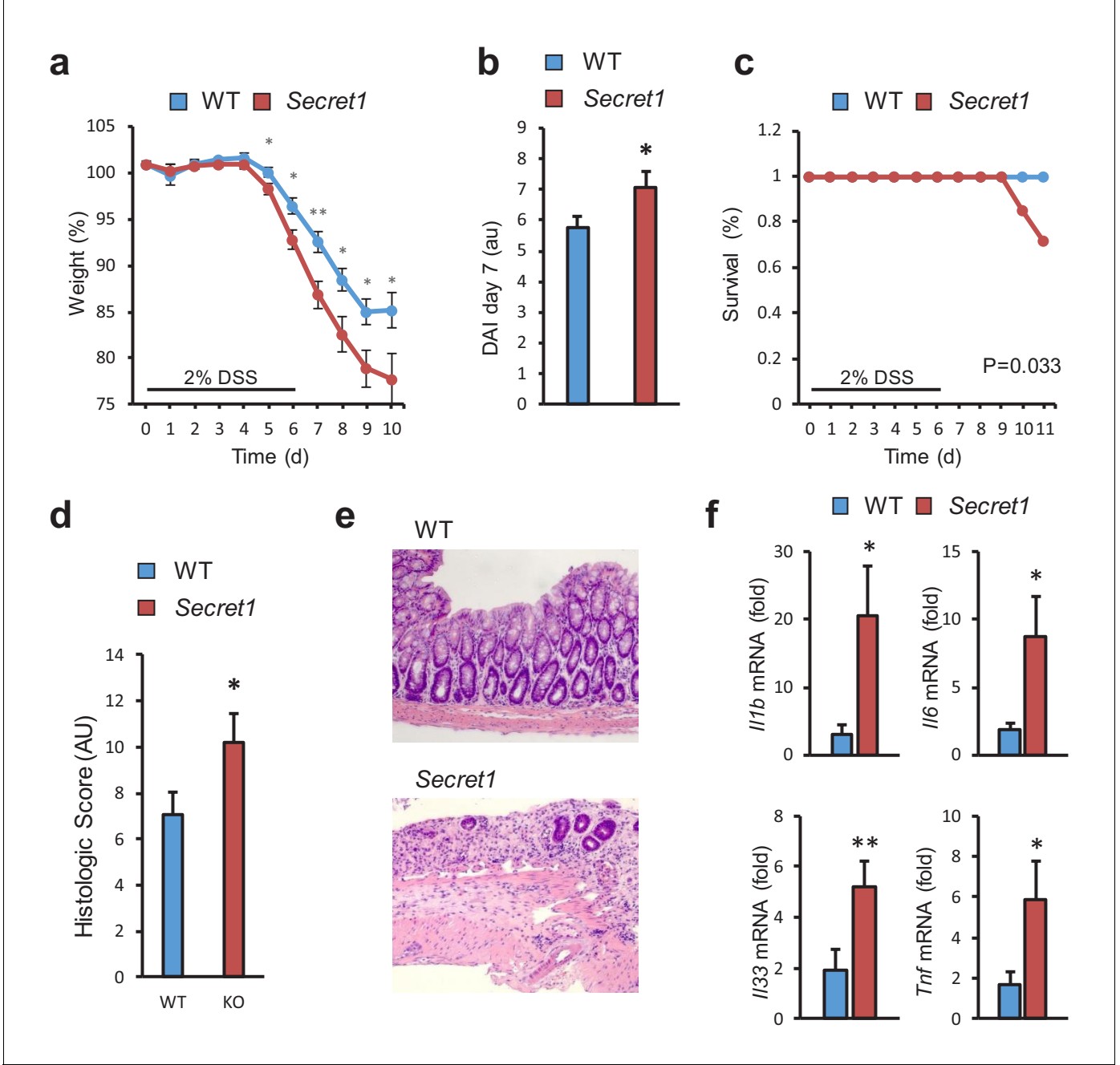

**Figure 5.** SCGN loss leads to increased sensitivity to DSS colitis. (a–c) Body weight, disease activity index (DAI), and survival of male WT (n = 13) and *Secret1* (n = 13) mice treated for 6 days with 2% DSS in their drinking water. (d) Histologic score of male and female WT (n = 11) and *Secret1* (n = 13) animals treated for 6 days with 2% DSS. (e) Representative microphotographs (20x) of colonic epithelium of DSS-treated mice. (f) Inflammatory gene expression measured by qRT-PCR from ceca of DSS-treated mice. Data in (a), (b) and (c) are representative of 2 experiments. *p<0.05. **p<0.01, two tailed unpaired *t* test in (a), (b) and (f). *p<0.05. one tailed unpaired *t* test in (d). S.E.M. was used for error bars in (a), (b), (d), (f). Log rank test was used in (c).

DOI: https://doi.org/10.7554/eLife.49910.025

The following source data and figure supplements are available for figure 5:

**Source data 1.** Source data for *Figure 5A*.
DOI: https://doi.org/10.7554/eLife.49910.032
**Source data 2.** Source data for *Figure 5B*.
DOI: https://doi.org/10.7554/eLife.49910.033
**Source data 3.** Source data for *Figure 5C*.

*Figure 5 continued on next page*

*Figure 5 continued*

DOI: https://doi.org/10.7554/eLife.49910.034

**Source data 4.** Source data for *Figure 5D*.

DOI: https://doi.org/10.7554/eLife.49910.035

**Source data 5.** Source data for *Figure 5F*.

DOI: https://doi.org/10.7554/eLife.49910.036

**Figure supplement 1.** Increased sensitivity to DSS in *Scgn* deficient animals.

DOI: https://doi.org/10.7554/eLife.49910.026

**Figure supplement 1—source data 1.** Source data for *Figure 5—figure supplement 1A*.

DOI: https://doi.org/10.7554/eLife.49910.027

**Figure supplement 1—source data 2.** Source data for *Figure 5—figure supplement 1B*.

DOI: https://doi.org/10.7554/eLife.49910.028

**Figure supplement 1—source data 3.** Source data for *Figure 5—figure supplement 1C*.

DOI: https://doi.org/10.7554/eLife.49910.029

**Figure supplement 2.** Immunophenotying of WT and *Secret1* mice.

DOI: https://doi.org/10.7554/eLife.49910.030

**Figure supplement 2—source data 1.** Source data for *Figure 5—figure supplement 2*.

DOI: https://doi.org/10.7554/eLife.49910.031

## Discussion

In this study, we identified a consanguineous family with recessive early-onset ulcerative colitis caused by a homozygous mutation in the *SCGN* gene. Various studies presented here confirm that the R77H substitution associated with the disease trait leads to loss of function for the encoded protein. Importantly, the p.R77H mutation identified in this family resulted in decrease hormone secretion and altered cellular localization of SCGN and its partner SNAP25 in vitro. In agreement with the notion that decreased function of SCGN can alter immune regulation in the intestine, we demonstrate that *Scgn*-deficient mice are more susceptible to DSS-induced colitis. Interestingly, targeting the colonic EEC compartment did not recapitulate the phenotype, suggesting that gut or central neurons are the likely relevant compartments that SCGN regulates. In aggregate, the genetic, cellular and mouse model results from this study implicate *SCGN* as a novel susceptibility gene for monogenic early-onset IBD.

In contrast to other forms of monogenic IBD, which frequently involve other significant phenotypes in addition to intestinal inflammation (e.g., immunodeficiency, gut developmental problems, etc.), our patients did not display any other clinically overt phenotypes outside of colitis. Interestingly, mutations in other regulators of the SNARE complex, namely STXBP2, can lead to altered NK cell degranulation, hemophagocytic lymphohistiocytosis and colitis (*Meeths et al., 2010*). Intriguingly, despite the restricted expression of *SCGN* in neuroendocrine lineages, there were no overt endocrinopathies or neurologic phenotypes. For example, in contrast to the age-dependent onset of diabetes mellitus previously reported in *Scgn*-deficient mice (*Malenczyk et al., 2017*), the fasting glycemia of these patients has been normal, which is remarkable considering that they have been exposed, at times, to glucocorticoids. Whether this reflects an intrinsic difference in the function of SCGN in mouse and human or is simply indicative of an age-related phenotype that is yet to manifest in these subjects remains to be seen. Similarly, the patients do not display any overt neurocognitive phenotypes, again in contrast with data from zebrafish and murine models of *Scgn* deficiency (*Deciphering Developmental Disorders Study, 2015*; *Hanics et al., 2017*). While the data from Zebrafish presented here indicate that SCGN[R77H] cannot rescue the developmental brain phenotype of deficient fish, it is still possible that the lack of overt neuroendocrine phenotypes may be attributable to *SCGN* p.R77H retaining some function compared to complete loss of SCGN expression, or may be indicative of species-specific phenotypes not evident in humans.

Unlike STXBP2, SCGN protein is absent from the immune system, pointing specifically to neuroendocrine cells as the likely source of pathology in these patients. It is important to note that both the EEC and neuronal compartments have been shown to be highly interconnected, with inputs reaching the central nervous system (*Kaelberer et al., 2018*). EECs play crucial roles in the hormonal control of satiety, digestion, energy metabolism and motility. A potential role for these cells in

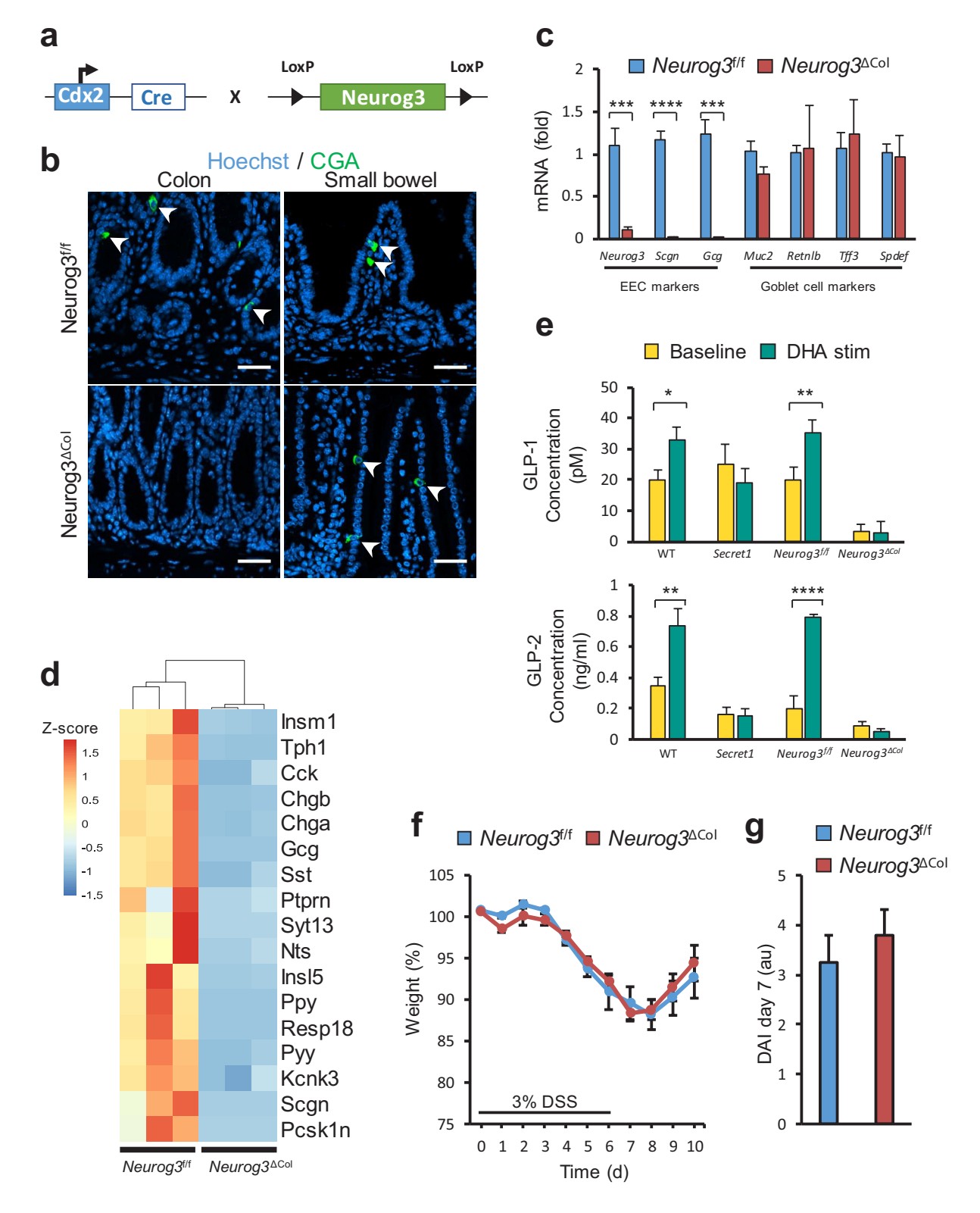

**Figure 6.** Loss of colonic EECs does not confer DSS susceptibility. (a) Diagram depicting the mating strategy used to generate colonic EEC (*Neurog3*^ΔCol) deficient mice. (b) CGA immunofluorescent staining of colon and small bowel from wild-type (*Neurog3*^f/f) and colonic EEC deficient (*Neurog3*^ΔCol) mice (c) qRT-PCR of epithelial lineage makers from colonic epithelium from *Neurog3*^f/f and *Neurog3*^ΔCol mice (n = 3 in each group). (d) Heat map presentation of top differentially expressed genes from RNA-seq of colonic epithelia of *Neurog3*^f/f and *Neurog3*^ΔCol mice. (e) Ex-vivo basal

*Figure 6 continued on next page*

*Figure 6 continued*

and DHA-stimulated GLP-1 and GLP-2 secretion from colonic explants of Secret1 and EEC deficient mice. (GLP-1: *Neurog3*$^{f/f}$ n = 7, *Neurog3*$^{\Delta Col}$ n = 6, WT n = 6, *Secret1* n = 6) (GLP2: *Neurog3*$^{f/f}$ n = 5, *Neurog3*$^{\Delta Col}$ n = 5, WT n = 6, *Secret1* n = 4). Bars represent the mean and error bars the S.E.M. (f–g) Body weight and DAI of conventionally raised *Neurog3*$^{f/f}$ and *Neurog3*$^{\Delta Col}$ mice treated for 6 days with 3% DSS (n = 15 in each group). Scale bar in (b) 50 μm. *p<0.05, **p<0.01, ***p<0.001 ****p<0.0001 two tailed unpaired *t* test in (c) and (e). S.E.M. was used for error bars in (c), (e), (f) and (g).
DOI: https://doi.org/10.7554/eLife.49910.037

The following source data is available for figure 6:

**Source data 1.** Source data for *Figure 6C*.
DOI: https://doi.org/10.7554/eLife.49910.038
**Source data 2.** Source data for *Figure 6D*.
DOI: https://doi.org/10.7554/eLife.49910.039
**Source data 3.** Source data for *Figure 6E*.
DOI: https://doi.org/10.7554/eLife.49910.040
**Source data 4.** Source data for *Figure 6F*.
DOI: https://doi.org/10.7554/eLife.49910.041
**Source data 5.** Source data for *Figure 6G*.
DOI: https://doi.org/10.7554/eLife.49910.042

immune regulation has been suggested by the fact that these cells express Toll-like receptors and other receptors for microbially-derived metabolites, which can stimulate hormone secretion by these cells (*Bogunovic et al., 2007*; *Lebrun et al., 2017*; *Palazzo et al., 2007*; *Selleri et al., 2008*). Nonetheless, our studies in mice with ablation of colonic EECs indicate that this is not the likely cellular compartment responsible for the observed intestinal pathology, and point to dysfunction of gut neurons or central nervous system neurons as possible culprits of the pro-inflammatory phenotype. In fact, a role for gut neurons in immune regulation has been previously demonstrated. For example, murine models of gut neuron and glial cell defects have been shown to predispose to experimental colitis (*Bush et al., 1998*; *Cheng et al., 2010*; *Fujimoto et al., 1988*), and gut neuron deficiency in humans, such as in Hirschsprung's disease, can be associated with perinatal severe colitis (*Demehri et al., 2013*; *Gosain, 2016*; *Gosain and Brinkman, 2015*). Future studies to dissect the contribution of specific cellular compartments will delineate additional mechanistic details for the immune dysregulation and pathology resulting from *SCGN* deficiency.

## Materials and methods

### DNA extraction, SNP arrays and linkage analysis

DNA was extracted from 5 mL of whole blood collected in EDTA coated tubes using an automated QIAGEN AUTOPURE LS – large sample nucleic acid purification instrument. DNA concentration and quality was assessed on a nanodrop instrument. DNA was aliquoted and stored at −20°C until use. DNA samples from probands and their unaffected siblings were genotyped for common variants along the genome using the Infinium HumanCoreExome Beadchip SNP array (Illumina), which includes 264,909 tag SNP markers (244,593 in exons). Genotyping was performed by the McDermott Center for Human Genetics according to the manufacturer's standard procedures. Areas of loss of heterozygosity that were shared by probands and not their unaffected siblings were identified as potential regions harboring a recessive allele responsible for disease linkage.

### Whole exome and whole genome sequencing

Three micrograms of genomic DNA isolated from peripheral blood were used for whole-exome sequencing. The exome was captured using the Agilent SureSelectXT Human All Exon V4 Kit and sequenced on the Illumina HiSeq 2500 generating 100 bp paired-end reads. Variant calls were filtered to only include exonic, non-synonymous changes with minor allele frequencies < 1% in ExAC, 1000 genomes and TOPMED. We filtered further to include variants that were exclusively shared by all probands in a homozygous manner and present at most in a heterozygote manner in unaffected siblings. The analysis was conducted both at the genome-wide level and subsequently, the variants were reanalyzed focusing on areas of LOH identified by genome-wide SNP arrays as noted above.

Because of gaps in WES capture affecting the areas of LOH, DNA samples from two probands were subsequently subjected to whole genome sequencing (Complete Genomics) and repeat analysis for rare variants (MAF <1%) in the segments of LOH that were shared between the two probands was performed. Furthermore, using WGS data, we remapped LOH segments. Heterozygous and homozygous variants were identified and assigned into 100 kb windows throughout the 6 p and 12q chromosome arms. Average heterozygosity was calculated for each window and plotted along the chromosome arm.

## Allelic discrimination assay

Samples from 2000 Hispanic individuals enrolled in the Dallas Heart study were used for direct genotyping for rs376721140 using TaqMan probes flanking the area of interest. The assays were performed by the McDermott Center for Human Genetics using an Applied Biosystems 7900HT real time PCR instrument.

## Plasmids

A plasmid containing a full-length copy of human *SCGN* cDNA was purchased from Open Biosystems (MHS6278-202826064). Forward and reverse primers containing sites for BamHI and NotI restriction enzymes respectively, were used to PCR amplify the coding sequence for *SCGN*. The product was digested with BamHI and NotI and ligated into the multicloning site of the pEBB vector. The inserted sequence (*SCGN*) was validated by Sanger sequencing. *SCGN* R77H was generated using a site-directed mutagenesis kit (Agilent), and the mutation was confirmed by Sanger sequencing. The coding sequences for *SCGN* (WT or R77H) were subcloned into the FG9 lentiviral expression system containing a N-terminus 2xHA tag, as described before (*Li et al., 2012*).

## Immunoblotting and immunoprecipitation studies

Cellular lysate preparation, protein electrophoresis, immunoblotting and immunoprecipitation was performed as previously described (*Burstein et al., 2004*). The antibodies used in our studies include SNAP25 (Abcam ab5666), SCGN (Santa Cruz biotechnology sc-374355), P84 (Genetex GTX70220), Actin (Sigma A5441), and anti-HA affinity matrix (Roche 11815016001).

## Cell culture and cell line generation

STC-1 cells were obtained from ATCC (Cat No. CRL-3254) and grown in culture using DMEM supplemented with 10% FBS. Identity testing was not performed as these cells were directly obtained from the vendor. Cells were tested and found to be Mycoplasma-free at time of experiments. To target the *Scgn* locus (exon 1), lentiviral CRISPR constructs (Lenti CRISPR system) were generated, as previously reported (*Shalem et al., 2014*). STC-1 cells were transduced with lentiviral particles and subjected to puromycin selection. Single clone isolation was carried out by limiting dilution, and clones were screened for loss of SCGN protein expression by western blot. Two independent clones were subsequently used for studies. SCGN protein expression was restored into these clones using lentiviral particles designed to express HA tagged WT or p.R77H human SCGN. Lentiviruses were generated using the FG9 system, with hygromycin resistance as a selection marker. Protein reexpression was confirmed by western blot.

## Cellular immunofluorescence staining and analysis

Immunofluorescence staining of STC-1 cells was carried out as previously described (*Phillips-Krawczak et al., 2015*). Briefly, cells were grown onto circular cover slips in 6-well plates. Once cells reached 70% confluency, cover slips were retrieved and cells were fixed with ice-cold fixing solution (4% paraformaldehyde in PBS) for 30 min. Cells were then permeabilized for 3 min using 0.15% Surfact-Amps (ThermoFisher) in PBS. The coverslips were then blocked for 30 min with blocking buffer (PBS and 3% goat serum – vector labs) and afterwards incubated overnight in primary antibody diluted in blocking buffer. Primary antibodies used for immunostaining included the following: Secretagogin (Santa Cruz biotechnology, sc-374355), SNAP25 (Abcam ab5666). After three washing steps in PBS, cells were incubated in secondary antibodies and respective fluorophores: goat anti-mouse alexa fluor 488 (Invitrogen) and goat anti-rabbit alexa fluor 555 (invitrogen). Images were obtained using a Nikon A1R confocal laser microscope system. The membranous and total cellular area plus

fluorescence intensity of individual cells was measured using the ImageJ software. Approximately 40 cells per cell-type were analyzed. A membranous to total cellular fluorescence intensity ratio was calculated.

### In-vitro GLP-1 release assay

All cells were counted and seeded on 6-well plates at equal density. Once cells were confluent they were placed in serum free media for a 2 hr period and washed twice with HEPES buffer (140 mM NaCl, 4.5 mM KCL, 20 mM HEPES, 1.2 mM $CaCl_2$, 1.2 mM $MgCl_2$, 20 mM Dextrose) before adding 400 µL of DHA-containing media. To prepare DHA-containing media, stock DHA (nu-chek U-84-A) solutions were prepared in 100% ethanol at a concentration of 10 mg/mL and frozen at −20°C until used. DHA-containing media at a concentration of 100 µM was prepared on the day of experiment by addition to a HEPES buffer, which was sonicated for 5 min prior to addition to cells. Cell supernatants were collected after 15 min and GLP-1 concentrations were determined using an ELISA kit (Millipore, EZGLPHS-35K). To account for possible differences in cellular density, ELISA results were normalized to total protein content (measured by Bradford assay) after cellular lysis of the corresponding wells.

### Zebrafish models

All zebrafish were kept in a standard aquatic facility, and raised at 28.5°C. Knockdown of *scgn* was achieved by injecting 5 ng of morpholino (GENE TOOLS) into 1 cell stage blastocysts, together with a tp53 MO to avoid the non-specific effects from injection (*Robu et al., 2007*). Morpholino efficiency was confirmed by RT-PCR and western blot analysis. For rescue experiments *scgn* MO was coinjected with mRNAs encoding either human SCGN wild-type or R77H into blastocysts of one-cell stage. Protein re-expression was confirmed by western blot (data not shown). In situ hybridization was carried out as previously described (*Luo et al., 2016*). Images were acquired using an OLYMPUS (SZX16) microscope. The midbrain was measured from a lateral view.

### Mouse strains

Mice were kept in a specific pathogen free (SPF) barrier facility, in a standard 12 hour day cycle and fed standard irradiated chow. *Scgn* deficient animals were generated at the UT Southwestern transgenic core by multiple zygote injection of custom synthetic sgRNA (Dharmacon) and Cas9 mRNA (Sigma). Mice were generated on a C57BL/6 background and resulting progeny was characterized for potential targeting of the *Scgn* allele by PCR amplification of a 214 bp region flanking the site of sgRNA cleavage. The amplified product was subjected to Sanger sequencing. Two founder mice with *Scgn* mutations resulting in significant deletions at this site, referred to here as *Secret1* and *Secret2* (*Figure 4—figure supplement 1a*), were selected for future studies. Genotyping of these mice was performed by amplification of this region and gel electrophoresis to detect the genomic deletions (*Figure 4—figure supplement 1b*).

Colonic EEC deficient animals were generated by mating the CDX2 Cre driver line (Jax 009350) against *Neurog3* floxed mice, a kind gift from Andrew Leiter with authorization from Gérard Gradwohl.

### DSS experiments

Acute colitis was induced by providing a 6 day course of DSS (Alfa aesar J63606-22) in drinking water at a concentration of 1%, 2% or 3% (g/dL, weight per volume). DSS solution was replaced every three days. On 10 day long DSS experiments, drinking water was replaced with autoclaved water as indicated on experimental figures. Mice of different genotypes were cohoused at the time of DSS experiments. Weight was recorded on a daily basis. Disease activity index (DAI) was calculated on day seven in a blinded fashion by adding individual scores from stool consistency, presence of blood in stool and weight change, as previously described (*Supplementary file 3*) (*Cooper et al., 1993*).

### Tissue staining

Tissue analysis from patient samples involved archived intestinal biopsy tissue blocks from control and affected individuals. Tissues from experimental animals were freshly isolated after euthanasia

and fixed overnight in freshly prepared 4% PFA, to be later embedded in paraffin. For all staining, 4 µm sections were utilized. Hematoxylin and eosin (HE) staining was performed by standard methods. For immunostaining, patient slides were stained using the Dako Omnis automated system, whereas slides obtained from experimental animals were manually stained. Briefly, the sections were first deparaffinized and rehydrated by serial xylene and decreasing alcohol concentration immersion. Antigen retrieval was performed by heat-induced epitope retrieval and citrate solution. Slides were incubated in blocking buffer for 45 min (PBS and 3% goat serum – vector labs), and then incubated overnight at 4°C in primary antibodies diluted in blocking buffer. Primary antibodies used for tissue immunostaining included the following: Secretagogin (Santa Cruz biotechnology sc-374355), chromogranin A (Abcam ab15160), chromogranin B (Abcam ab12242), 5-HT (Immunostar 20080), SNAP25 (Abcam ac5666), GCG (Santa Cruz biotechnology sc-514592), Tuj1 (Biolegend 801213) and synaptophysin (Abcam ab32127). Slides were washed in PBS three times (5 min each), and then incubated for 30 min in secondary antibodies in blocking buffer. Secondary antibodies used included goat anti-mouse alexa fluor 488 (Invitrogen) and goat anti-rabbit alexa fluor 555 (invitrogen). Slides were then washed in PBS on two occasions and then incubated for 10 min in PBS and Hoechst at 1:10,000 dilution. Slides were washed in PBS on three additional occasions prior to mounting using gold antifade reagent (Invitrogen).

## Tissue histological analysis

For DSS colitis histological severity scoring, HE slides were reviewed by a GI pathologist in a blinded fashion. Tissues were scored using a modified scoring system (*Supplementary file 2*) generated from combining the total score calculated from Neurath's DSS score (*Wirtz et al., 2017*) and multiplying it by involvement score as suggested by Cooper (*Cooper et al., 1993*). Morphometric analysis of SCGN and CGA intestinal expression in the mouse was performed by staining tissues from four adult wild-type animals. Images (at least 13 per mouse and per region) were obtained using an epifluorescence microscope (Zeiss AxioObserver epifluorescence microscope). Images were scored for SCGN and CGA positivity by two independent observers and an average calculated from both quantifications.

## Tissue mRNA expression

For DSS experiments, cecal tissue was collected at the end of the experiments and immediately stored in RNA later solution (QIAGEN), following the manufacturer's instructions. RNA stabilized tissue was later thawed and 20 mg of tissue was disrupted using a glass tissue douncer and homogenized using QIAshredder columns prior to RNA extraction using RNeasy spin columns (QIAGEN). For preparation of small bowel and colonic epithelial isolates, the entire small bowel or colon were carefully excised and intestinal contents rinsed with KRB buffer (10 mM D-glucose, 0.5 mM $MgCl_2$, 4.6 mM KCl, 120 mM NaCl, 0.7 mM $Na_2HPO4$, 1.5 mM $NaH_2PO4$, 15 mM $NaHCO_3$). The intestinal segments were then filled with 1 mL KRB buffer containing 10 mM EDTA and 1 mM DTT and tied off on either end. These segments were placed in a sealed 50 mL conical tube containing 30 mL of KRB buffer and placed in a rocking deck incubator at 220 RPM for 40 min at 37°C. The luminal contents were collected into 1.5 mL microfuge tubes and washed three times with KRB buffer prior to storing in RNAlater solution at −80°C. RNA was then extracted following instructions by the manufacturer.

We performed reverse transcription using 3–5 µg of RNA and the superscript III system (Invitrogen). We carried out quantitative real time PCR using SYBR green mix. Primers used for *Il6*, *Il1b*, *Il33*, *Tnf*, *Lyz*, *Neurog3*, *Muc2*, *Retnlb*, *Tff3*, *Spdef* and *Scgn* are provided in *Supplementary file 4*. Relative transcript levels were calculated using the ΔΔCt method.

## RNA sequencing

RNA Seq was performed as previously reported (*Starokadomskyy et al., 2016*). Briefly, RNA was extracted from epithelial preparations with RNeasy columns (QIAGEN) as per instructions from the manufacturer. RNA integrity was determined using a bioanalyzer. Library preparation was performed at the UTSW microarray core using the TruSeq RNA sample preparation kit. Sequencing was performed on an Illumina platform HiSeq2500 sequencer. Sequencing data were analyzed using a custom workflow designed by the UTSW high performance computing cluster (BioHPC). Briefly, after adapter trimming, sequences were aligned to reference genome, GRCm38, using Hisat2. Duplicate

reads were marked and removed using SAMBAMBA. Features were then counted using feature-counts. Differential expression analysis was performed using edgeR with cutoffs of 2 and 0.05 for FC and FDR respectively.

## Colonic GLP-1 and GLP-2 release assay

Adult animals (WT or *Secret1*) were euthanized and the colon was immediately excised. Intestinal contents rinsed with ice-cold incubation buffer (138 mM NaCl, 5.6 mM KCl, 2.6 mM CaCl$_2$, 1.2 mM MgCl$_2$, 4.2 mM NaHCO$_3$, 1.2 mM NaH$_2$PO$_4$, 10 mM HEPES), as previously described (*Lebrun et al., 2017*). The colon was opened along the mesenteric axis, and rolled open. Colons from individual mice were placed in a 6-well plate containing 3000 µL of dextrose-free and DHA-free incubation buffer. They were incubated for 10 min at room temperature and then at 37°C, 5% CO2 for an additional 10 min. A total of 200 µL of incubation buffer was collected for baseline GLP-1 secretion analysis. The colon was then gently tapped dry and re-placed in a new well with 3 mL of incubation buffer with 10 mM dextrose and 100 µM DHA (this solution was prepared fresh and sonicated for 3 min prior to use). At 60 min, 200 µL of buffer was collected for analysis. PMSF was added at a concentration of 100 µM (as DPP IV inhibitor) to each timed collection. A GLP-1 (Millipore EZGLPHS-35K) or GLP-2 (Crystal Chem 81514) ELISA kit was used for quantitative GLP-1 analysis according to manufacturer's instructions.

## Microbiota analysis by 16S sequencing

Morning fecal pellets were collected from co-housed WT and *Secret1* adult animals aged 10–12 weeks. Fecal pellets were immediately frozen in liquid nitrogen and stored until use. DNA was extracted using QIAGEN power fecal kit according to manufacturer's instructions. DNA purity and concentration were measured on a nanodrop device. Paired-end 16 s sequencing was carried out by a commercial vendor (SeqMatic) using an Illumina MiSeq platform. Sequencing data were then subjected to standard QIIME2 pipeline workflow (*Bokulich et al., 2018*; *Caporaso et al., 2010*), consisting of pre-processing quality preparation (trimming, demultiplexing, DADA2 quality filtering), phylogenetic profiling, alpha and beta diversity analysis, sequence alignment, taxonomic assignment and distribution analysis; and differential abundance testing using ANCOM.

## Flow cytometry

Lamina propria (LP) cells were isolated as described before (*Kathania et al., 2016*). In brief, colons were flushed to wash off fecal content and opened longitudinally. Colons were then cut into 0.5 cm pieces, transferred to flasks and shaken for 25 min at 37°C in HBSS containing 5 mM EDTA and 10 mM HEPES supplemented with 10% FBS. Cell suspensions were passed through a cell strainer. The remaining colonic tissue was washed with cold PBS, minced, transferred to conical flasks, and shaken for 25 min at 37°C in DMEM containing 0.25 mg/ml VII collagenase, 0.125 U/ml LiberaseTM, 10 mM HEPES pH 8, 0.1 M CaCl2, 0.05% DNase1, and 10% FBS. Cell suspensions were collected and passed through a strainer before staining and analysis. Cells were washed and then incubated with Fc block (553142, BD Bioscience). Subsequently, cells were stained with combinations of antibodies. Antibodies used were Lin-V450 (51–9006958, BD Biosciences), CD45.2-APC (17-0454-81, eBioscience), NK-p46- FITC (11-3351-82, eBioscience) IL-23R-PE (FAB16861P- R and D), CD4-FITC (100406, Biolegend), CD11c-PE-Cy7 (558079, BD Biosciences), Ly6G-eFlour-450 (48-5931-82, eBioscience), CD19-V450 (560375, BD Biosciences), NK1.1 FITC (553164, BD Biosciences), CD11b-FITC (553310, BD Biosciences) and CD25-PE (12-0251-81, eBioscience). Viability staining was done by 7-AAD (420404, Biolegend). Data were acquired with a FACSCanto II (BD) and analyzed with FlowJo software (Tree Star).

## Statistical analysis

The unpaired Student's *t* test was utilized to parametric variables between two groups. The log rank test was calculated for statistical analysis of survival rates. Statistical analysis of 16S sequencing was performed as described above. For zebrafish experiments statistical analyses were performed using one-way ANOVA, Tukey's multiple comparisons test incorporated in Prism 7 (GraphPad Software).

## Study approval

All human studies were carried out in accordance with UT Southwestern Medical Center institutional review board guidelines under an approved protocol (STU 112010–130). All subjects agreed to participation and written informed consent was obtained from all participants or legal guardians. Assent was obtained from individuals older than 10 years of age at time of enrollment. Murine studies were approved by the UT Southwestern Institutional Animal Care and Use Committee under study number APN 102011. All zebrafish (Danio rerio) experiments were performed according to standard procedures, and were performed in accordance with the guidelines of the animal ethical committee of Sichuan University.

## Acknowledgements

We wish to thank patients and families involved in this study. We also acknowledge the help from John Shelton and the staff at UTSW's molecular pathology core, the Children's Health histology core with help with immunostaining, and the McDermott Center Bioinformatics Core facility. We also thank the McDermott Center Human Gene Discovery laboratory. We thank the transgenic core at UTSW for their help in generating animal models. We also want to thank Jonathan Cohen and Helen Hobbs at the McDermott Center for their invaluable comments throughout this project and for their support with genotyping of subjects in the DHS for the *SCGN* variants.

This work was supported by the Pollock Family Center for Research in IBD, as well as generous gifts from Crowley Family Foundation, the McCune Family Foundation and the Lewis Family Foundation to EB. LS-D was supported through NIH 5 K12 HD-068369–05 and Children's Health Clinical Research Advisory Committee: CCRAC 195. The work of LAB is supported by NIH R01 DK105068, and the work of JJR was supported by NIH grant UL1TR001105. DJ is supported by Natural Science Foundation of China (NSFC) grants (#91854121, #31871429).

## Additional information

### Funding

| Funder | Grant reference number | Author |
| --- | --- | --- |
| Eunice Kennedy Shriver National Institute of Child Health and Human Development | 5 K12 HD-068369-05 | Luis F Sifuentes-Dominguez |
| Children's Health Clinical Research Advisory Committee | 195 | Luis F Sifuentes-Dominguez |
| National Institute of Diabetes and Digestive and Kidney Diseases | DK105068 | Linda A Baker |
| National Center for Advancing Translational Sciences | UL1TR001105 | Jonathan J Rios |
| National Natural Science Foundation of China | 91854121 | Da Jia |
| UT Southwestern | Pollock Family Center for Research in Inflammatory BowelDisease | Ezra Burstein |

The funders had no role in study design, data collection and interpretation, or the decision to submit the work for publication.

### Author contributions

Luis F Sifuentes-Dominguez, Conceptualization, Data curation, Formal analysis, Funding acquisition, Investigation, Methodology, Writing—original draft, Project administration, Writing—review and editing, Performed most of the cellular, molecular, immunofluorescence and animal experiments, and wrote manuscript; Haiying Li, Investigation, Methodology, Aided with cloning; Ernesto Llano, Shuai Tan, Investigation, Aided with animal experiments; Zhe Liu, Investigation, Methodology,

Performed zebrafish experiments; Amika Singla, Investigation, Contributed with cellular secretion studies; Ashish S Patel, Investigation, Gathered clinical data; Mahesh Kathania, Data curation, Formal analysis, Investigation, Methodology, Performed immunophenotyping of colonic lamina propria cells; Areen Khoury, Project administration, Aided with patient enrollment; Nicholas Norris, Investigation, Aided with protein binding experiments; Jonathan J Rios, Data curation, Formal analysis, Investigation, Performed genetic analyses; Petro Starokadomskyy, Investigation, Helped with image acquisition and quantification; Jason Y Park, Data curation, Performed human histological analyses; Purva Gopal, Formal analysis, Investigation, Performed DSS histological scoring; Qi Liu, Investigation, Methodology, Aided with cellular immunofluorescence; Lillienne Chan, Formal analysis, Investigation, Methodology, Project administration, Gathered clinical data; Theodora Ross, Resources, Investigation, Performed genetic analyses; Steven Harrison, Formal analysis, Performed genetic analyses; K Venuprasad, Formal analysis, Investigation, Methodology, Performed immunophenotyping of colonic lamina propria cells; Linda A Baker, Funding acquisition, Investigation, Methodology, Performed genetic analyses; Da Jia, Conceptualization, Formal analysis, Supervision, Funding acquisition, Investigation, Methodology, Writing—review and editing, Supervised zebrafish experiments and helped with protein binding experiments; Ezra Burstein, Conceptualization, Data curation, Formal analysis, Supervision, Funding acquisition, Investigation, Visualization, Methodology, Writing—original draft, Project administration, Writing—review and editing, Supervised most of the cellular, molecular, immunofluorescence, genetic and animal experiments, Developed experimental strategy and wrote this manuscript

## Author ORCIDs
Luis F Sifuentes-Dominguez (iD) https://orcid.org/0000-0002-9525-071X
Linda A Baker (iD) https://orcid.org/0000-0001-8272-4886
Ezra Burstein (iD) https://orcid.org/0000-0003-4341-6367

## Ethics
Human subjects: All human studies were carried out in accordance with UT Southwestern Medical Center institutional review board guidelines under an approved protocol (STU 112010-130). All subjects agreed to participation and written informed consent was obtained from all participants or legal guardians. Assent was obtained from individuals older than 10 years of age at time of enrollment.

Animal experimentation: Murine studies were approved by the UT Southwestern Institutional Animal Care and Use Committee under study number APN 102011. All zebrafish (Danio rerio) experiments were performed according to standard procedures, and were performed in accordance with the guidelines of the animal ethical committee of Sichuan University.

## Decision letter and Author response
Decision letter https://doi.org/10.7554/eLife.49910.052
Author response https://doi.org/10.7554/eLife.49910.053

# Additional files

## Supplementary files
• Supplementary file 1. Copy number variation and loss of heterozygosity (LOH) analysis by SNP array.
DOI: https://doi.org/10.7554/eLife.49910.043

• Supplementary file 2. Scoring system for inflammation-associated histological changes in the colon (DSS).
DOI: https://doi.org/10.7554/eLife.49910.044

• Supplementary file 3. Disease activity index for colitis model.
DOI: https://doi.org/10.7554/eLife.49910.045

• Supplementary file 4. Primer Sequences.
DOI: https://doi.org/10.7554/eLife.49910.046

• Supplementary file 5. Key resource table.
DOI: https://doi.org/10.7554/eLife.49910.047

• Transparent reporting form DOI: https://doi.org/10.7554/eLife.49910.048

## Data availability

Sequencing data have been deposited in GEO under accession code GSE134202. Data generated during this study is included in the manuscript.

The following dataset was generated:

| Author(s) | Year | Dataset title | Dataset URL | Database and Identifier |
|---|---|---|---|---|
| Sifuentes-Dominguez LF, Burstein E | 2019 | Transcriptome-wide gene-expression analysis of colonic epithelium from enteroendocrine cell-deficient mice | https://www.ncbi.nlm.nih.gov/geo/query/acc.cgi?acc=GSE134202 | NCBI Gene Expression Omnibus, GSE134202 |

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
