## [Decision Letter]

**Acceptance summary:**

We are grateful that you have chosen to publish your data with us, showing that a rare secretagogin variant (SCGN: a calcium sensor that is expressed in intestinal enteroendocrine cells and gut neurons) variant is linked to colitis and proctitis in sib offspring from consanguineous partners. Your investigations with SCGN variants and lentiviral complementation in a cell line that secretes GLP-1 has shown the that the variant has hypomorphic activity and your genetic survey shows that it affects an EF hand motif, consistent with calcium-dependent signaling. Cellular staining and immunoprecipitation results show the defective association with SNAP25 and cellular localization, especially on the plasma membrane. Your in vivo work demonstrates that the SCGN enteroendocrine cells are predominantly in the large intestine, and that murine SCGN deletion increases the susceptibility to dextran sulfate sodium-induced colonic injury and inflammation.

**Decision letter after peer review:**

Thank you for submitting your article "SCGN deficiency results in colitis susceptibility" for consideration by *eLife*. Your article has been reviewed by three peer reviewers, one of whom is a member of our Board of Reviewing Editors, and the evaluation has been overseen by Wendy Garrett as the Senior Editor. The following individual involved in review of your submission has agreed to reveal their identity: Holm H Uhlig (Reviewer #3).

The reviewers have discussed the reviews with one another and the Reviewing Editor has drafted this decision to help you prepare a revised submission.

Summary:

This study shows that a rare secretagogin (SCGN) variant is linked to colitis and proctitis in three sib offspring from consanguineous partners. Further work with SCGN variants and lentiviral complementation in a cell line the secretes GLP-1 has shown the hypomorphism of the variant and a genetic survey shows that this is placed in a EF hand motif, consistent with calcium-dependent signaling. Cellular staining and immunoprecipitation results demonstrate the defective association with SNAP25 and cellular localization, especially on the plasma membrane.

The SCGN enteroendocrine cells are shown to be predominantly in the large intestine, and that murine SCGN deletion increases the susceptibility to DSS colitis although deficiency of murine colonic enteroendocrine cells were unable to phenocopy the susceptibility to colitis, so the mechanisms of pro-inflammatory functional SCGN-deficiency and specifically whether neuronal specific SCGN-deficiency promotes colitis in vivo remain unclear.

Essential revisions:

The reviewers have reached the conclusion that in vivo studies of neuronal specific SCGN deficiency would be beyond the scope of the current manuscript, although any further progress in this direction would be especially welcomed.

1) The Abstract should be revised since this is not just a susceptibility gene but a causative monogenic defect with Mendelian inheritance. Also, the Abstract currently says that the SCGN deficiency recapitulates 'colitis susceptibility'. This is strictly true, but the fact that DSS needs to be used as a trigger is very relevant to a correct impression of the result.

2) Table 1. shows areas of shared loss-of-heterozygosity (LOH) among affected probands as defined by SNP array – the filter strategy should be provided in more detail and variants listed.

3) The authors should screen the different databases whether any other predicted homozygous essential loss of function variant (stop codon/frameshift) is present in public databases. Are any deletions described with neurodevelopment defects in databases such as Clinvar?

4) It would be helpful to have some more patient details with the follow-up. Since the phenotype can apparently present with proctitis, it would be important to understand the severity and trajectory of the condition.

5) Immunoprecipitation of SCGN in STC-1 cells expressing either wild type or the R77H variants show comparable co-precipitation of endogenous SNAP25 whereas recombinant human SCGNR77H expressed in bacteria displayed reduced SNAP25 in vitro binding – what is the mechanistic basis for this difference? Differential glycosylation?

6) Since GLP-2 has been more widely reported to have anti-inflammatory immune properties, could this be the explanation of the SCGN effect (Frontiers in Immunology 2017 01734)?

---

## [Author Response]

Essential revisions:The reviewers have reached the conclusion that in vivo studies of neuronal specific SCGN deficiency would be beyond the scope of the current manuscript, although any further progress in this direction would be especially welcomed.1) The Abstract should be revised since this is not just a susceptibility gene but a causative monogenic defect with Mendelian inheritance. Also, the Abstract currently says that the SCGN deficiency recapitulates 'colitis susceptibility'. This is strictly true, but the fact that DSS needs to be used as a trigger is very relevant to a correct impression of the result.

We have amended the Abstract to reflect the comments above. Please note the changes in the revised manuscript (highlighted in gray).

2) Table 1. shows areas of shared loss-of-heterozygosity (LOH) among affected probands as defined by SNP array – the filter strategy should be provided in more detail and variants listed.

We have added a table (Supplementary file 1) that lists all copy number variants and LOH regions identified in all 5 individuals tested (P1, P2, P3, S1, S2). Further, we have explained in more detail in the Results section how this data was filtered in order to identify areas of shared LOH among probands (subsection “A very rare variant in SCGN causes inherited early-onset ulcerative colitis”, second paragraph). We have also included an analysis for LOH obtained from WGS data from P1 and P2 that confirmed the LOH segments noted by SNP arrays (see Figure 1—figure supplement 2).

3) The authors should screen the different databases whether any other predicted homozygous essential loss of function variant (stop codon/frameshift) is present in public databases. Are any deletions described with neurodevelopment defects in databases such as Clinvar?

We have performed this suggested analysis and include it now in the paper. For ease of review, we copy that content here:

“Further screening of different databases (ExAC, ClinVar) indicates that there are no individuals identified thus far with predicted homozygous essential loss of function variant (stop codon/frameshift). […] In fact, all variants of SCGN reported in ClinVar are large copy number variants involving dozens to hundreds of genes, with phenotypes involving congenital anomalies and intellectual disabilities (Accession numbers VCV000608768, VCV000608767, VCV000608764, VCV000443497, VCV000443496, VCV000155430, VCV000150044, VCV000149747).”

4) It would be helpful to have some more patient details with the follow-up. Since the phenotype can apparently present with proctitis, it would be important to understand the severity and trajectory of the condition.

We have added an additional figure that summarizes the clinical trajectory and course of the disease over time in all affected probands (Figure 1—figure supplement 1).

5) Immunoprecipitation of SCGN in STC-1 cells expressing either wild type or the R77H variants show comparable co-precipitation of endogenous SNAP25 whereas recombinant human SCGNR77H expressed in bacteria displayed reduced SNAP25 in vitro binding – what is the mechanistic basis for this difference? Differential glycosylation?

To try to address this comment, we repeated the in vitro binding assay using HA-SCGN expressed in human cells (HEK293T). Unlike *E. coli*-generated recombinant protein which binds poorly to SNAP-25, HEK293T-generated protein has minimal to no change in binding ability in pulldown assays using STC-1 cellular lysates as a source of SNAP-25 (Author response image 1).

**Author response image 1. respfig1:** HEK293T cells, which do not express SNAP25, were transfected with HA-tagged SCGN (WT or R77H). These proteins were subsequently purified through an HA binding resin. The purified proteins, immobilized on the resin were then incubated with cellular lysates of STC-1 cells (0.75mg of lysate offered), and SNAP25 binding was determined by immunoblotting. Three technical replicates of the IP were performed (top three panels on the left). Input from HEK293 cells are shown on the right. This experiment is representative of 3 independent iterations.

It therefore suggests that there is likely a mammalian modification that may be at play as suggested by the reviewer. Additional work in collaboration with Dr. Da Jia (co-author in this paper) has led to the crystal structure of the SCGN-SNAP25 complex, which is the subject of a separate report. This structure now indicates that the interaction between these proteins is based on a different domain of SCGN, and not EF hand 2 where the R77 residue is located. Therefore, on balance, the decreased binding of recombinant, bacterially made SCGN^R77H^ is currently hard to explain structurally. After consideration of the various results, we have decided that for this paper it would be most appropriate to present the clear mislocalization phenotype and leave the in vitro binding data out of this paper until it can be more clearly interpreted.

6) Since GLP-2 has been more widely reported to have anti-inflammatory immune properties, could this be the explanation of the SCGN effect (Frontiers in Immunology 2017 01734)?

We appreciate this remark and have performed additional colonic ex-vivo GLP-2 secretion assays to answer this question. Our new data (Figure 6E) indicate that GLP-1 and GLP-2 secretion are both limited in *Scgn* deficient animals. Importantly, deletion of all EECs in the colon of *Neurog3*^ΔCol^ animals leads to as much or more significant defects in GLP-1 and GLP-2 secretion. Since these EEC depleted mice do not have increased sensitivity to DSS colitis, we conclude that defective GLP-2 secretion cannot explain the colitis sensitivity of the *Scgn* knockout.